# Metabolic modeling reveals determinants of prebiotic and probiotic treatment efficacy across multiple human intervention trials

**Nick Quinn-Bohmann**[1], **Alex V. Carr**[1], **Sean M. Gibbons**[1,2,3,4,5]*

1 Institute for Systems Biology, Seattle, Washington, United States of America, 2 Department of Bioengineering, University of Washington, Seattle, Washington, United States of America, 3 Department of Genome Sciences, University of Washington, Seattle, Washington, United States of America, 4 eScience Institute, University of Washington, Seattle, Washington, United States of America, 5 Department of Environmental and Occupational Health Sciences, University of Washington, Seattle, Washington, United States of America

* sgibbons@isbscience.org

## Abstract

Prebiotic, probiotic, and combined (synbiotic) interventions often show variable outcomes across individuals, driven by complex interactions between introduced biotics, the endogenous microbiota, and the host diet. Predicting individual-specific success or failure of probiotic and prebiotic therapies remains a major challenge. Here, we leverage microbial community-scale metabolic models (MCMMs) to predict probiotic engraftment and microbiota-mediated short-chain fatty acid (SCFA) production in response to probiotic and prebiotic interventions. Using data from two human clinical trial cohorts, testing a five-strain probiotic combined with the prebiotic inulin designed to improve metabolic health and an eight-strain probiotic designed to treat recurrent *Clostridioides difficile* infections, respectively, we show that MCMM-predicted engraftment largely agrees with measurements, achieving 75%–80% accuracy. Engraftment probabilities varied across taxa. MCMMs captured treatment-driven shifts in predicted SCFA production, and higher model-predicted growth rates of *Akkermansia muciniphila* were negatively associated with glucose area under the curve (AUC) in the first trial, providing clues about the mechanisms underlying treatment efficacy. Extending these models to a third human cohort undergoing a healthy diet and lifestyle intervention revealed substantial inter-individual variability in predicted responses to increasing dietary fiber, which were significantly associated with baseline-to-follow-up changes in cardiometabolic health markers. Finally, our simulation results suggested that personalized prebiotic selection may further enhance probiotic efficacy. Together, these findings demonstrate the potential of metabolic modeling to guide personalized microbiome-mediated interventions.

provided the original author and source are credited.

**Data availability statement:** Shotgun metagenomic sequencing data and metadata from Validation Study A (clinical trial NCT03893422) are not publicly posted but are available on reasonable request from the original investigators. Inquiries to access the data from Validation Study A can be made at john.eid@ pendulum.co and will be responded to within 10 business days. qPCR data for probiotics strains and measures of glucose AUC can be found in S1 and S3 Data, respectively. These data can also be found at the associated GitHub repository (https://github.com/ wholebiome/NCT03893422/tree/master/ BMC_Microbio_2022_10.1186_s12866-021-02415-8). Shotgun metagenomic sequencing data and metadata from Validation Study B are publicly available through the NCBI Sequence Read Archive (SRA) under accession number PRJNA755324. 16S rRNA amplicon sequencing data for the Arivale cohort can be found on the NCBI Sequence Read Archive under accession numbers PRJNA826530 and PRJNA826648. Clinical lab values for corresponding participants are available in S4 Data. All underlying data for results described in Fig 1 can be found in S1 Data. All underlying data for results described in Figs 2, S1, and S2 can be found in S2 Data. All underlying data for results described in Fig 3 can be found in S3 Data. All underlying data for results described in Fig 4 can be found in S4 Data. All underlying data for results described in Figs 5, S3, S4 and S5 can be found in S5 Data. All code and analyses from this study can be found on Zenodo under DOI: https://doi.org/10.5281/zenodo.18037976.

**Funding:** This study was funded, in part, by a research grant from Pendulum (to SMG), the manufacturer of the synbiotic tested in Perraudeau et al. (2020). This work was also supported by the National Institute of Diabetes and Digestive and Kidney Diseases (NIDDK) of the National Institutes of Health (NIH) under award number R01DK133468, and by a Global Grants for Gut Health Award from Nature Portfolio and Yakult (to SMG). The funders had no role in the study design, data collection and analysis, decision to publish, or preparation of the manuscript.

**Competing interests:** I have read the journal's policy and the authors of this manuscript have the following competing interests: SMG is a

## Introduction

The human colonic microbiota plays a critical role in regulating host physiology and immunity [1–5]. Probiotic, prebiotic, and synbiotic (i.e., combination of prebiotics and probiotics) administration has been shown to influence the metabolic outputs of the endogenous microbiota, inhibit pathogen colonization, and enhance mucosal barrier integrity [6,7]. Emerging evidence suggests that synbiotic interventions may even impact brain function via the gut-brain axis [8–10].

Despite these promising findings, the efficacy of probiotic, prebiotic, and synbiotic interventions can vary widely between individuals, posing a major challenge for their widespread clinical application [11,12]. Engraftment of a probiotic strain—the ability of the administered microbe to grow and persist within the gut ecosystem—depends on multiple factors, including the availability of a suitable metabolic niche, competitive and cooperative interactions with endogenous microbiota, and interactions with the host immune system [13,14]. Beyond the composition of the commensal microbiota, the interplay between factors like total commensal biomass, propagule pressure (i.e., the probiotic dose), available prebiotic substrates, and host diet can impact engraftment [15]. In order to rationally predict probiotic engraftment and prebiotic effects across individuals and dietary backgrounds, we require methods that integrate this complexity.

Genome-scale metabolic models (GEMs) provide a powerful mechanistic framework for estimating microbial growth and metabolism [16,17]. Recent advances in constraint-based modeling have extended this approach to diverse microbial communities, yielding microbial community-scale metabolic models (MCMMs) [18]. By integrating genome-scale metabolic reconstructions from hundreds of gut bacterial taxa with flux balance analysis, MCMMs enable predictions of microbe-microbe interactions, competition for resources, and ecosystem-scale metabolic behaviors [18,19]. These models have demonstrated predictive accuracy in estimating microbial growth rates, short-chain fatty acid (SCFA) production, methane production, and metabolic shifts in response to disease status or dietary interventions [18–22]. Notably, prior work has shown that MCMMs can capture the engraftment potential of opportunistic pathogens, such as *Clostridioides difficile*, suggesting that similar approaches could be used to model probiotic engraftment in individual hosts [23].

To validate this approach, we aimed to use MCMMs to predict the enrichment of probiotic species based on data from two human intervention trials. First, we leveraged data from a study conducted by Perradeau and colleagues (Validation Study A) [24]. In this double-blind, placebo-controlled trial, participants previously diagnosed with type 2 diabetes mellitus were treated for 12 weeks with either a placebo or one of two multi-strain synbiotic cocktails (WBF-010 and WBF-011). WBF-010 contained *Bifidobacterium infantis*, *Clostridium beijerinckii*, and *Clostridium butyricum*, supplemented with a low dose of inulin (0.3g), a prebiotic fiber known to support probiotic growth. WBF-011 contained the same prebiotic and probiotic contents as WBF-010, but with the addition of two probiotic strains: *Akkermansia muciniphila* and *Anaerobutyricum hallii*. Participants who received the WBF-011 synbiotic exhibited a significantly greater reduction in the glucose area under the curve (AUC) during a standard

paid member of the scientific advisory board for Thorne. This work is unrelated to Thorne, and Thorne had no involvement in the study design, data collection and analysis, decision to publish, or manuscript preparation.

**Abbreviations:** AUC, area under the curve; Ct, cycle threshold; GEMs, genome-scale metabolic models;MCMMs, microbial community-scale metabolic models; qPCR, quantitative PCR; SCFA, short-chain fatty acid.

glucose tolerance test from Week 0 to Week 12, compared to the placebo group, indicating a beneficial population-level effect on glycemic control [24]. For this analysis, only data from participants in the WBF-011 treatment group were used, given that WBF-010 did not show a significant clinical effect on the primary outcome variables. In the second study (Validation Study B) by Dsouza and colleagues, participants were given an 8-strain probiotic cocktail (VE303) designed to treat recurrent *Clostridioides difficile* infections [12]. VE303 consisted of *Enterocloster bolteae*, *Anaerotruncus colihominis*, *Sellimonas intestinalis*, *Clostridium symbiosum*, *Blautia* sp001304935, *Dorea longicatena*, *Longicatena innocuum*, and *Flavonifractor plautii*, though *Blautia* sp001304935 and *Longicatena innocuum* were omitted from the current analysis due to a lack of available GEMs in the AGORA database [25].

Using metagenomic sequencing data, quantitative PCR (qPCR), and immune and metabolic profiling, we assessed the ability of MCMMs to accurately predict probiotic engraftment and the subsequent impact on host metabolism and immune function. Specifically, we evaluated how well MCMM-derived predictions align with observed shifts in microbial composition and clinical markers. Finally, we expand MCMM simulations to a cross-sectional cohort of generally healthy individuals from the Arivale cohort who received a healthy diet and lifestyle intervention, where they increased their exercise and intake of dietary fiber. MCMM-predicted increases in butyrate production after switching from a standard European to a high-fiber diet were significantly associated with longitudinal changes in cardiometabolic health markers in this intervention cohort. Using a subset of this Arivale cohort, we explored personalized prebiotic, dietary, and probiotic interventions optimized to increase butyrate or propionate production. Together, these results highlight a promising computational platform for designing personalized prebiotic, probiotic, and synbiotic interventions for human cohorts.

## Results

### Validation Study A: qPCR shows enrichment of probiotic species in the treatment group

Abundances of focal microbial species were determined using qPCR (see Methods).

*A. muciniphila* and *B. infantis* exhibited significant enrichment from Week 0 to Week 12 across most participants (Fig 1A; paired *t* test, *A. muciniphila*: $p = 3.2 \times 10^{-4}$; 1-sample *t* test, *B. infantis*: $p = 4.8 \times 10^{-6}$). In contrast, *C. beijerinckii*, *A. hallii*, and *C. butyricum* showed no comparable enrichment. These trends suggest short-term engraftment of *A. muciniphila* and *B. infantis* in the majority of treated individuals. Binarized engraftment scores were defined based on qPCR amplification thresholds (see Methods; S1 Data).

### Validation Study A: MCMMs predict species-level enrichment

Models were constructed from baseline taxonomic assignments ($N = 21$) to assess whether MCMMs could predict the engraftment patterns observed in the qPCR results (Fig 1A; see Methods). For each individual, models simulating the addition of 5 probiotic species were used to predict engraftment patterns observed in Week 12 qPCR data (see Methods).

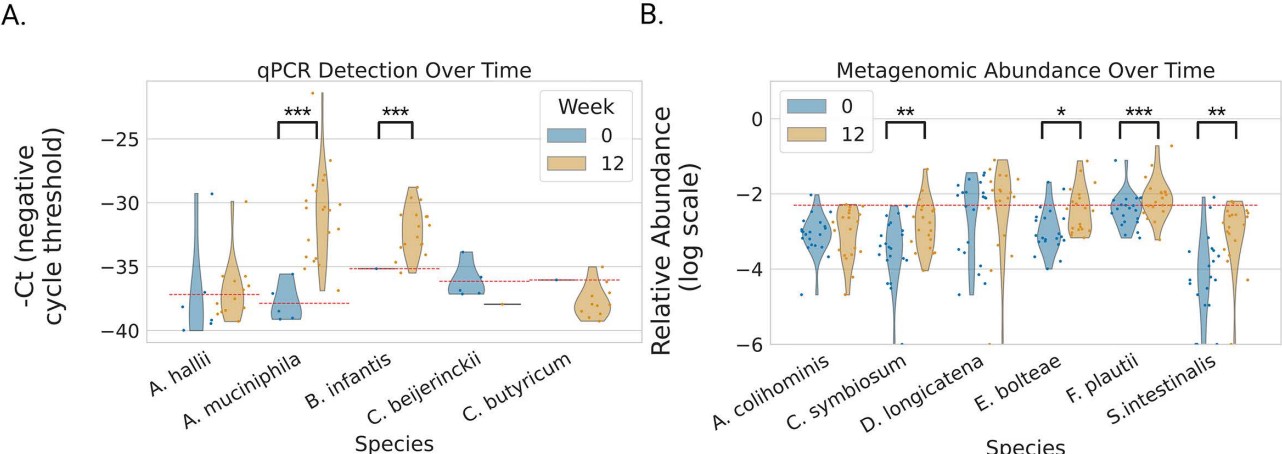

**Fig 1. Probiotic strain abundance quantification in two probiotic intervention trials. (A)** Based on qPCR data, there was a significant enrichment in *Akkermansia muciniphila* and *Bifidobacterium infantis* probiotics in Validation Study A at Week 12 compared to Week 0. There was no enrichment of *Clostridium beijerinckii*, *Clostridium butyricum*, and *Anaerobutyricum hallii* probiotics. **(B)** Based on shotgun metagenomic data, there was significant enrichment in *Enterocloster bolteae*, *Flavonifractor plautii*, *Sellimonas intestinalis*, and *Clostridium symbiosum* in Validation Study B at Week 12 compared to Week 0. Only *E. bolteae* and *F. plautii* showed notable enrichment above the relative abundance threshold of 0.005. Significance determined by paired *t* test between timepoints, or one-sample *t* test when only a single datapoint was available, * = $p < 0.05$, ** = $p < 0.01$, *** = $p < 0.001$. Underlying available in S1 Data.

Across all comparisons (5 predictions for each of 21 samples, total $N = 105$), model-predicted growth and binarized enrichment scores showed 84.7% agreement (Fig 2A, Cohen's $\kappa = 0.68$, indicating substantial agreement). *C. beijerinckii* was correctly predicted as not growing for all 21 participants, and *A. muciniphila* was correctly predicted as growing in all 21 participants. *B. infantis* had the lowest accuracy, with incorrect prediction in 8 of 21 cases. Overall, there was significant concordance between predictions and observations (Fig 2B). Across all strains, 58 instances of growth were correctly predicted, 31 instances of non-growth were correctly predicted, 3 cases were incorrectly predicted as growth, and 13 cases were incorrectly predicted as non-growth (Fisher's exact test, $p = 4.4 \times 10^{-13}$). Full results are found in S2 Data.

### Prediction of microbial short-chain fatty acid production

To evaluate the predicted functional impact of probiotic administration, we analyzed the model-predicted production of butyrate and propionate, two key SCFAs with well-documented metabolic health-promoting, anti-inflammatory, and gut barrier-strengthening properties [26]. Prior work has demonstrated the accuracy of MCMM-based predictions for SCFA production in the human gut microbiome [19], providing a strong rationale for employing this approach to assess the mechanistic contributions of both the probiotic and prebiotic components of the WBF-011.

To systematically dissect the influence of probiotic and prebiotic interventions on SCFA production, we compared the production of propionate and butyrate between the baseline condition and the synbiotic condition. Due to the original study using a minimal prebiotic dose (0.3 g of inulin), we also simulated the addition of a larger prebiotic dose (30 g of inulin). Following simulation, total MCMM-predicted butyrate and propionate production rates were compared across conditions (Fig 3A–3B).

As expected, the no treatment group exhibited relatively low levels of butyrate production (Fig 3A, No Treatment: 14.25 ± 2.27 mmol/gDW/h). The addition of the probiotic cocktail with a minimal prebiotic did not significantly increase the level of butyrate produced by the community (Fig 3A, WBF-011 + Inulin, 0.3 g: 14.81 ± 3.72 mmol/gDW/h). In a simulated synbiotic intervention with a larger dose of inulin, a significant increase was observed (Fig 3A, WBF-011 + Inulin, 30 g: 23.61 ± 2.99 mmol/gDW/h, Mann–Whitney *U* test, $p = 6.1 \times 10^{-3}$). Data for butyrate predictions can be found in S3 Data.

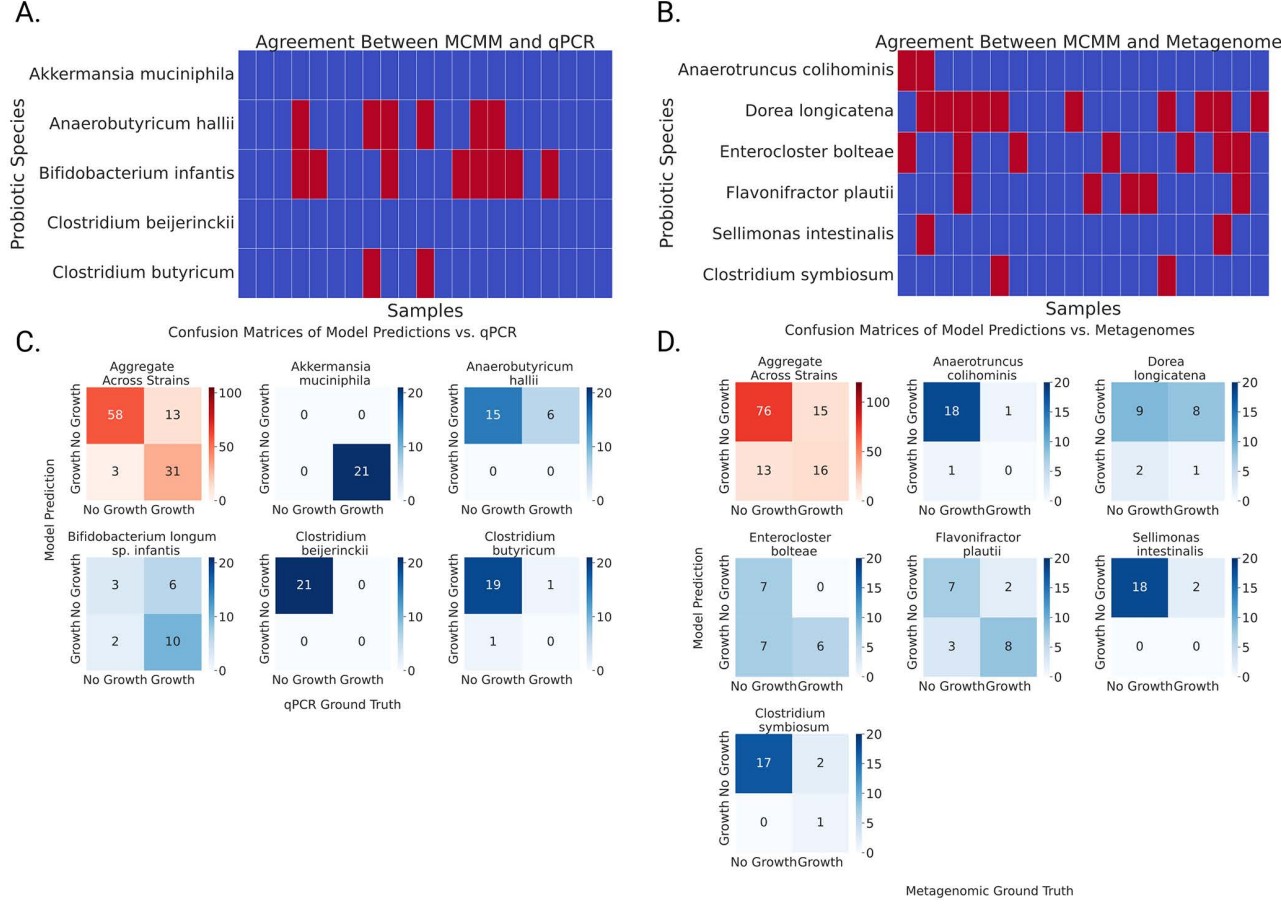

**Fig 2. MCMM predictions for probiotic growth show significant agreement with observed engraftment across two human intervention trials.** **(A)** Binarized MCMM-predicted growth agreed with qPCR engraftment scores in 89 out of 105 observations (Cohen's κ = 0.68, indicating moderate agreement). Blue boxes indicate agreement, red boxes indicate disagreement between the model and the qPCR data. **(B)** Confusion matrices describing results across and within probiotic strains, showing highly significant agreement (Fisher's Exact Test, aggregate results across all strains, p = 4.4x10⁻¹³). **(C)** Binarized MCMM-predicted growth predictions agree with metagenomic engraftment scores in 92 out of 120 observations (Cohen's κ = 0.38, indicating fair agreement). Blue boxes indicate agreement, red boxes indicated disagreement between the model and the qPCR data. **(D)** Confusion matrices describing results across and within probiotic strains, showing highly significant agreement (Fisher's Exact Test, aggregate results across all strains, p = 1.3x10⁻⁴). Underlying available in S2 Data.

Similar results were observed for propionate. The no treatment, and probiotic + minimal prebiotic group showed low levels of propionate production (Fig 3B, No Treatment: 55.92 ± 5.72 mmol/gDW/h; WBF-011 + Inulin, 0.3 g: 45.24 ± 3.30 mmol/gDW/h). The higher dose prebiotic combination significantly increased the production of propionate (Fig 3B, WBF-011 + Inulin, 30 g: 114.78 ± 5.66 mmol/gDW/h, Mann–Whitney $U$ test, $p = 9.3 \times 10^{-7}$). Predicted propionate results can be found in S3 Data.

### Glucose AUC related to predicted *Akkermansia muciniphila* growth rate

In Perraudeau and colleagues [27], glucose AUC was a primary metabolic endpoint. A significant difference in the change in glucose AUC from Week 0 to Week 12 (ΔglucoseAUC) was observed between participants receiving WBF-011 and those receiving the placebo (Fig 3C; Mann–Whitney $U$ test, $p = 5.0 \times 10^{-3}$). This effect was not observed for the WBF-010 formulation, which lacked *A. muciniphila* and *A. hallii* [24]. Combined with the absence of a significant WBF-011-induced

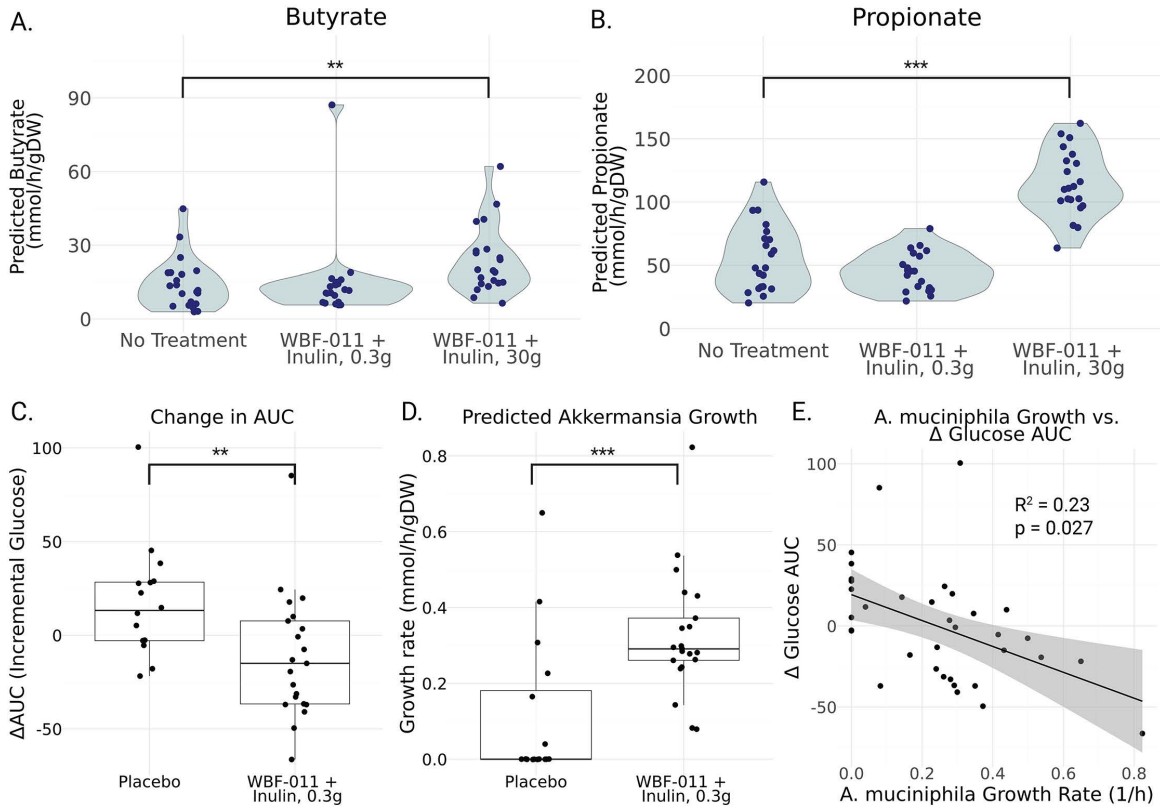

**Fig 3. MCMM-predicted SCFA production rates shift in response to synbiotic treatment and *Akkermansia muciniphila* growth rates are associated with clinical responses in Validation Study A. (A)** Butyrate production did not increase with the probiotic treatment when combined with a minimal prebiotic dose. Combining the probiotic treatment with a larger prebiotic dose results in a significant increase in butyrate production. **(B)** Propionate production also does not show significant changes with the probiotic treatment when combined with a minimal prebiotic dose. A more substantial prebiotic dose results in a significant increase in propionate production. Each point represents the predicted production level for a single sample. Significance was determined by Mann–Whitney $U$ test **: $p < 0.01$, ***: $p < 0.001$. **(C)** Participants in the WBF-011 treatment group showed a significantly larger decrease in glucose AUC between Week 0 and Week 12 than those in the placebo group. Significance was determined by Mann–Whitney $U$ test **: $p < 0.01$. **(D)** *A. muciniphila* showed a significantly higher predicted growth rate in the WBF-011 treatment group than the placebo group. Significance was determined by Mann–Whitney $U$ test ***: $p < 0.001$. **(E)** Predicted growth rate of *A. muciniphila* displays a negative association with the change in glucose AUC (Δglucose AUC) between Week 0 and Week 12 in the WBF-011 treatment group. Each point represents a single sample, the black line represents a least squares regression between Δglucose AUC and predicted growth rate, and the shaded gray region represents the 95% confidence interval. Underlying available in S3 Data.

change in MCMM-predicted butyrate or propionate production reported above and the generally low predicted engraftment of *A. hallii*, we hypothesized that *A. muciniphila* growth rate may be associated with clinical responsiveness. Indeed, the MCMM-predicted growth rate of *A. muciniphila* was significantly higher in the WBF-011 group compared to the placebo group (Fig 3D; Mann–Whitney $U$ test, $p = 9.4 \times 10^{-4}$). Furthermore, we found that ΔglucoseAUC was significantly negatively associated with the predicted growth rate of *A. muciniphila* within the WBF-011 treatment group (Fig 3E; linear regression, $R^2 = 0.23$, $p = 0.027$). Data for glucose AUC and predicted *A. muciniphila* growth rates can be found in S3 Data.

## Validation Study B: Metagenomic data indicate enrichment of probiotic species

Microbial relative abundances of focal species were determined using shotgun metagenomics (see Methods). *Entercloster bolteae* and *Flavonifractor plautii* showed an average enrichment following treatment, from Week 0 to Week 12 (Fig 1B; paired $t$ test, *E. bolteae*: $p = 5.5 \times 10^{-3}$; *F. plautii*: $p = 4.7 \times 10^{-2}$). *Sellimonas intestinalis* and *Clostridium symbiosum* also

showed significant enrichment ([Fig 1B](); paired *t* test, *S. intestinalis*: $p = 8.4 \times 10^{-4}$; *C. symbiosum*: $p = 7.4 \times 10^{-3}$), but stayed below a relative abundance of 0.005 in nearly all cases. *D. longicatena* also showed abundance above 0.005 in most cases, but it did not show a significant increase between timepoints. Binarized engraftment scores were assigned based on increases in relative abundance between timepoints, and being above a minimal relative abundance threshold of 0.005 (see [Methods](); [S1 Data]()).

## Validation Study B: MCMMs predict species-level enrichment

Models for Validation Study B were constructed using baseline taxonomic profiles ($N = 20$). For each individual, we modeled the addition of 6 of the 8 VE303 species; the remaining two were omitted because GEMs were unavailable. These models were used to predict the engraftment patterns observed in the Week 12 metagenomic sequencing data (see [Methods]()).

Across all comparisons (6 predictions for each of 20 samples, total $N = 120$) there was 76.7% agreement between model-predicted growth and binarized enrichment scores ([Fig 2C](), Cohen's $\kappa = 0.38$, indicating fair agreement). *A. colihominis*, *S. intestinalis*, and *C. symbiosum* were the best predicted species, with only two incorrect predictions each. *D. longicatena* was poorly predicted, with 10 incorrect predictions. There was significant agreement between predictions and observations ([Fig 2D]()). Across all observations, 16 instances were correctly predicted as growth, 76 were correctly predicted as non-growth, 13 instances were incorrectly predicted as growth, and 15 were incorrectly predicted as non-growth (Fisher's exact test, $p = 1.3 \times 10^{-4}$). The Cohen's $\kappa$ score was lower than in Validation Study A, largely due to the imbalance between the two observed engraftment states (31 instances of growth and 89 instances of non-growth), which inflates the expected chance agreement and depresses $\kappa$. For this reason, $\kappa$ downweights the otherwise strong (76.7%) concordance between MCMMs and the metagenomic data. Full results are found in [S2 data]().

## MCMM predictions link dietary fiber response to clinical health markers

To evaluate whether MCMM-predicted prebiotic- and probiotic-responsive SCFAs were associated with changes in host health, we modeled metabolic outputs from microbiome samples collected from 1,786 individuals enrolled in a former precision-wellness program with paired 16S amplicon sequencing data, blood-based clinical chemistries, and longitudinal follow-up data available (Arivale cohort) [27]. Arivale participants were coached to increase their exercise and to increase their fruit and vegetable intake. Each sample was simulated under two dietary conditions: a low-fiber standard European diet (i.e., a rough proxy for the baseline, pre-intervention diet) and a high-fiber diet enriched in resistant starch (i.e., a rough proxy for the post-intervention diet). Across participants, simulations predicted heterogeneous increases in butyrate production when transitioning from the low- to the high-fiber intervention ([Fig 4A]()). Regression analyses revealed that larger predicted increases in butyrate (Δbutyrate) were significantly associated with improvements in multiple cardiometabolic markers, including lower LP-IR, fasting insulin, and HOMA-IR ([Fig 4B]()). Negative associations were also observed between Δbutyrate and immune activation markers, such as total white blood cell count, neutrophils, and lymphocytes. A full set of predicted butyrate, clinical labs, and regression results is found in [S4 Data]().

## MCMM-optimized prebiotic and probiotic interventions designed to improve personalized probiotic engraftment and SCFA production profiles

To assess the potential power of our MCMM approach, we applied participant-specific simulations to a sub-cohort of Arivale participants with available fecal shotgun metagenomic sequencing data ($N = 156$), which tend to provide slightly more accurate constraints for MCMMs than 16S amplicon data [19].

Within the context of this Arivale sub-cohort, interventions that included the WBF-011 strains combined with eight different prebiotic and dietary contexts were simulated: (1) a standard European diet without prebiotic supplementation, (2) a standard European diet supplemented with inulin, (3) a standard European diet supplemented with pectin, (4) a

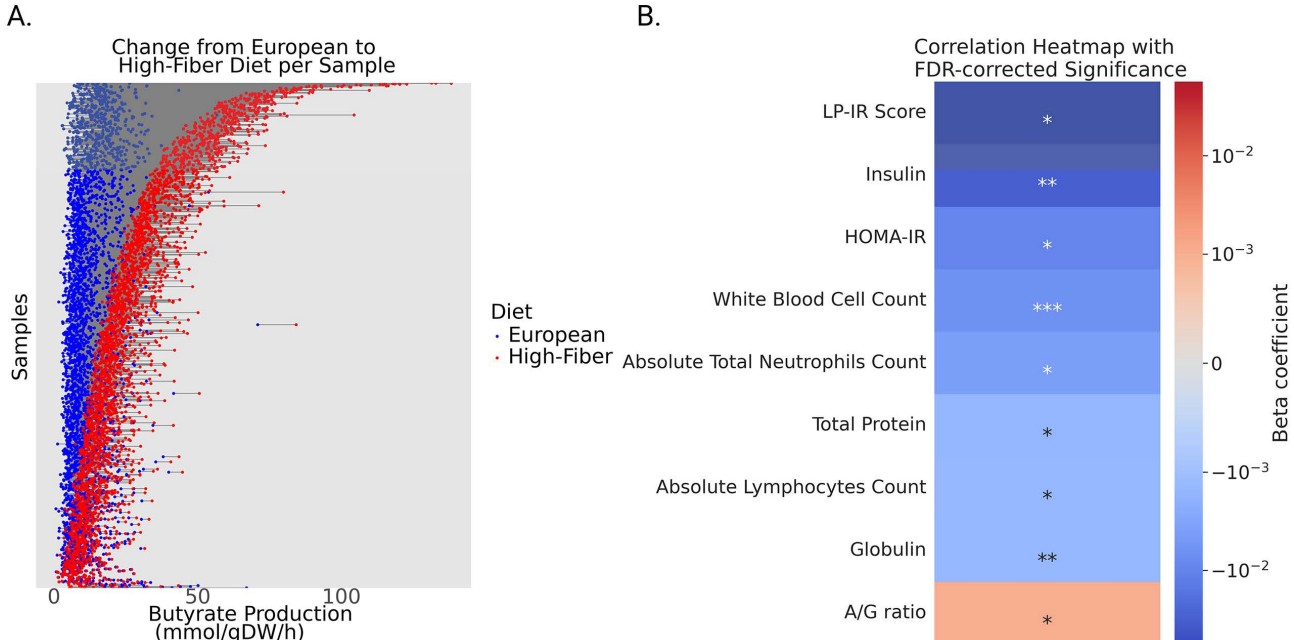

**Fig 4. Switching from a low- to high-fiber diet in the Arivale cohort causes non-uniform shifts in MCMM-predicted butyrate production, which were significantly associated with longitudinal changes in several clinical health markers. (A)** Simulating a dietary shift from a low-fiber stan-dard European diet (blue points) to a high-fiber diet (red points) results in a non-uniform increase in butyrate production across the study population ($N = 1,786$). Points from the same individual are connected by a horizontal gray line. **(B)** Change in butyrate production between diets (Δbutyrate) cor-relates with change in values of clinical chemistries across over a 6–12 month timespan, during a healthy lifestyle coaching program. Associations were determined using multiple regression adjusting for age, sex, and baseline values for each respective chemistry. Correction for multiple comparisons was performed using the Benjamini-Hochberg method. *: $p < 0.05$, **: $p < 0.01$, ***: $p < 0.001$. Underlying available in S4 Data.

standard European diet supplemented with resistant starch, (5) a standard European diet supplemented with maltodextrin, (6) a standard European diet supplemented with psyllium husk, (7) a standard European diet supplemented with hemp seed, and (8) a standard high-fiber diet. The addition of the probiotic consortia from Validation Study A showed signifi-cant decreases in predicted butyrate in every case except the no prebiotic and hemp seed contexts, where a significant increase was observed (Fig 5A, Mann–Whitney $U$ test, $p < 0.05$). Similarly, predicted propionate production decreased significantly when the probiotic cocktail was added, except in the no prebiotic or hemp seed contexts, where no signifi-cant differences were observed (Fig 5B, Mann–Whitney $U$ test, $p < 0.05$). Predicted acetate production showed a signif-icant increase when the probiotic was added in most cases, except in the no prebiotic and hemp seed contexts, where a significant decrease was observed, and in the starch context, where no significant difference was observed (Fig 5C, Mann–Whitney $U$ test, $p < 0.05$). Lactate production showed a significant increase when the probiotic cocktail was added in concert with pectin, psyllium husk, hemp seed, or in the no prebiotic context (Fig 5D, Mann–Whitney $U$ test, $p < 0.05$). To assess whether changes in predicted SCFA production were driven primarily by increases in overall biomass, we repeated these analyses using SCFA flux normalized by the predicted community growth rate. The normalized results (S3 Fig) displayed patterns consistent with the original analysis (Fig 5A–5D). Probiotic engraftment predictions for the WBF-011 cocktail varied only slightly across prebiotic and dietary backgrounds, indicating that endogenous microbiota composition and probiotic identity were stronger drivers of engraftment outcomes than prebiotic and dietary contexts (Fig 5E).

For each individual, the optimal prebiotic–probiotic combination on the standard European diet was identified, in terms of maximizing predicted butyrate or propionate production rates (Fig 5F and 5G). While some patterns emerged, the opti-mal intervention varied across individuals (S4 and S5 Figs). For butyrate and propionate production, psyllium husk with no

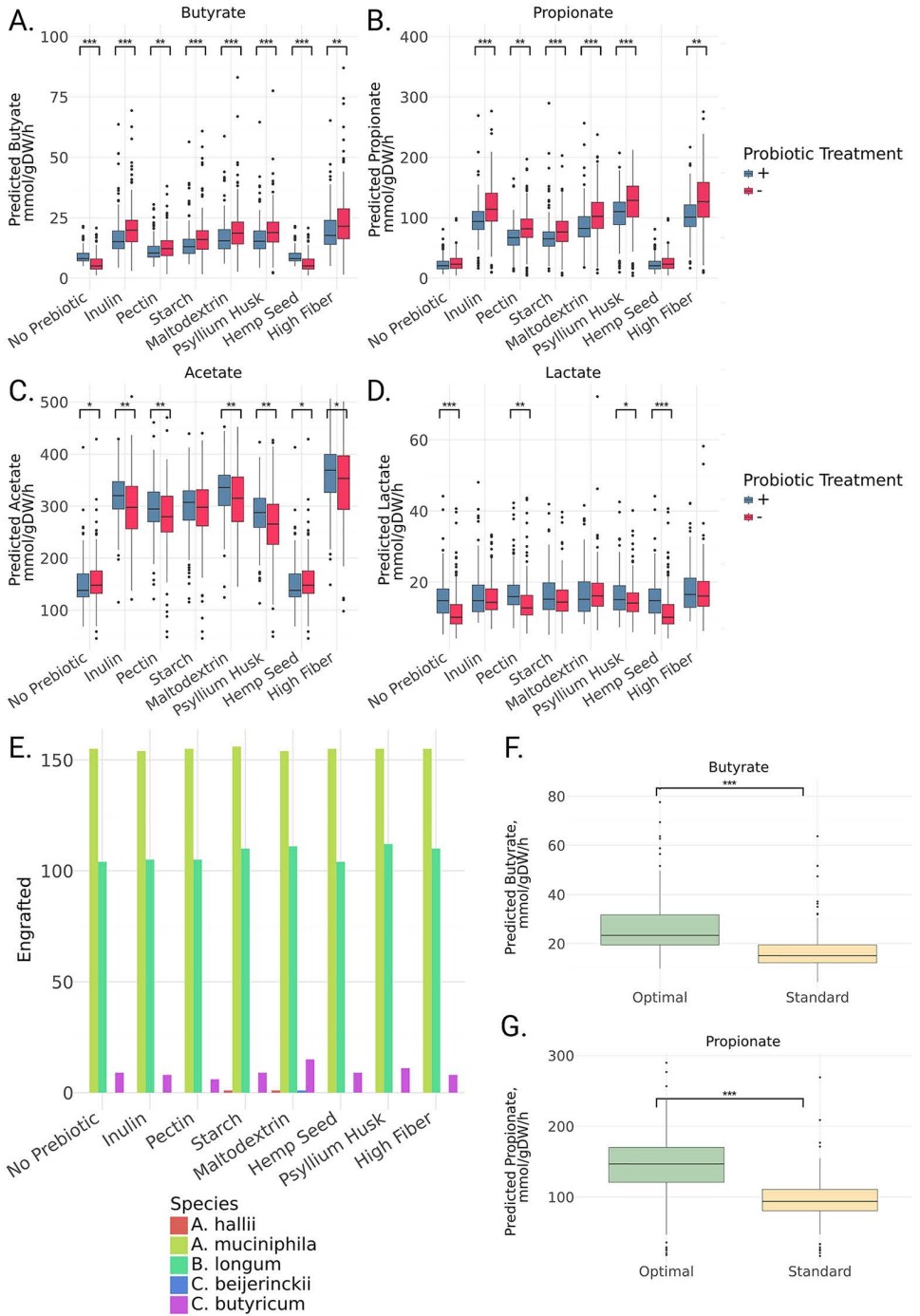

**Fig 5. MCMM-predicted SCFA and lactate production rates and WBF-011 probiotic species engraftment across simulated probiotic and prebiotic interventions in a cohort of generally healthy Americans. (A–D)** Predicted butyrate, propionate, acetate, and lactate production rates were calculated upon treatment with prebiotic substrates on a standard European diet (inulin, pectin, resistant starch, maltodextrin, psyllium, or hemp seed), or with a shift from a standard European to a standard high-fiber diet. Addition of a probiotic cocktail further shifted production rates, and tended to decrease butyrate and propionate production rates, relative to the prebiotic-alone treatments. Acetate and lactate production were generally higher in cases where the probiotic cocktail was added. Color encoding demonstrates the effect of adding a probiotic cocktail to the prebiotic/dietary treatments. Significance between the no probiotic and probiotic contexts determined by Mann–Whitney $U$ test, $* = p < 0.05$, $** = p < 0.01$, $*** = p < 0.001$. **(E)** Predicted engraftment of probiotic species, using a growth rate binarization threshold of $10^{-3}$, calculated for each prebiotic addition. In all cases, *A. muciniphila* and

*B. infantis* showed success in greater than 100 samples, while *A. hallii, C. beijerinckii,* and *C. butyricum* did not engraft in more than 15 samples at most. **(F, G)** For both predicted butyrate and propionate production, an optimal prebiotic and/or probiotic treatment combination was selected and compared against a standard treatment, which consisted of inulin and the 5-strain probiotic cocktail. In both cases, the individual-specific optimal treatment showed significantly higher levels of production. Significance determined by Mann–Whitney *U* test, *** = *p* < 0.001. Underlying available in S5 Data.

probiotic cocktail was the most common optimal intervention. However, many individuals showed a synbiotic combination as their optimal intervention (S4 and S5 Figs). Overall, a personalized optimal solution resulted in significantly higher predicted levels of propionate and butyrate production when compared with the "standard of care" synbiotic treatment used in the original WBF-011 trial (i.e., combining inulin with the five-strain probiotic cocktail).

Results of probiotic addition in the Arivale cohort can be found in S5 Data.

## Discussion

The ability of orally administered probiotics to successfully engraft in the gut is influenced by multiple ecological and host-specific factors, including the pre-existing composition of the gut microbiome and the background diet. A method for the rational prediction of probiotic engraftment and health-relevant microbial metabolite production, in the context of a given microbiota and diet, would open up new avenues for precision prebiotic, probiotic, and synbiotic design. Prior work by our group has demonstrated that our MCMM platform can be leveraged to predict personalized SCFA production rates and personalized pathobiont engraftment risk [19,23]. In order to validate our MCMM methodology in the context of synbiotic design, we used existing data from Validation Studies A and B to determine whether or not we can retrospectively predict relevant outcomes. Both studies showed variable engraftment rates across probiotic organisms, along with variable clinical responses. Given this heterogeneity, these trial data represented ideal test cases for our MCMM predictions.

Assessing empirical engraftment—by measuring relative enrichment of a probiotic taxon post-treatment via qPCR or metagenomics, and comparing these to MCMM predictions—is challenging. The presence of endogenous strains of the same species can make the detection of probiotic strains from metagenomic data challenging. Species-level enrichment of probiotic taxa can be difficult to distinguish from increases in the abundances of related strains already present. In Validation Study A, strain-specific qPCR primers often yielded no baseline detection, so post-treatment detection above a species-specific threshold was attributed to the administered strains. In Validation Study B, where baseline metagenomes contained endogenous strains of the same species, engraftment was defined by a 2-fold criterion: detection above a global threshold and an increase relative to the individual baseline. Although these metrics do not prove long-term engraftment, they provide a rough proxy for short-term enrichment/growth of the administered probiotic species.

Despite the inherent complexity of predicting probiotic growth, the MCMM-based framework demonstrated substantial accuracy, showing over 84% agreement with empirical probiotic engraftment metrics in Validation Study A (Fig 2A) and over 75% agreement in Validation Study B (Fig 2C). These models incorporate key ecological determinants, like endogenous microbiome composition and the availability of prebiotic and dietary substrates, to estimate probiotic growth rates. Notably, the high level of accuracy in engraftment predictions was achieved in spite of the fact that we lacked personalized constraints on dietary intake (i.e., a standard European diet was applied to each model). It is also worth noting the instances where the model failed—for instance, *B. infantis* and *D. longicatena* were poorly predicted in Validation Study A and Validation Study B, respectively. This could be the result of unannotated pathways in the GEMs of these poorly-predicted taxa, or a mismatch between the *in silico* dietary constraints applied in the model and the actual consumption patterns of study participants.

A fundamental goal of probiotic and prebiotic supplementation is to modulate the production of key microbial metabolites that support host health. Among these, SCFAs—such as butyrate and propionate—are particularly significant due to their roles in maintaining gut barrier integrity, regulating proper immune function, and supporting metabolic homeostasis [19,26]. Synbiotic administration provides a means of enhancing SCFA production, and MCMMs can be leveraged to

predict these metabolic shifts. Notably, our analysis revealed a synergistic effect between prebiotic administration and the WBF-011 probiotic cocktail. The probiotic cocktail paired with only a small amount of inulin showed a minimal effect with regard to butyrate and propionate production (Fig 3A and 3B). However, increasing the amount of inulin supplementation resulted in a significant increase in the production of both butyrate and propionate (Fig 3A and 3B).

Given accurate predictions of probiotic engraftment, we sought to identify associations between probiotic administration and the primary outcomes of the initial study. Perraudeau and colleagues reported a significant reduction in postprandial glucose AUC over the 12-week trial in the WBF-011 treatment group (Fig 3C), but not in the placebo or WBF-010 groups [27]. The primary difference between WBF-011 and WBF-010 was the inclusion of two additional strains: *A. muciniphila* and *A. hallii*. While *A. hallii* was rarely predicted to grow, *A. muciniphila* was consistently predicted to grow in most individuals receiving WBF-011 (Fig 3D). We hypothesized that *A. muciniphila* growth might be linked to changes in glucose AUC. Consistent with this hypothesis, MCMM-predicted growth rates of *A. muciniphila* were negatively associated with ΔglucoseAUC both across the WBF-011 and placebo groups and within the WBF-011 group (Fig 3C–3E). These findings support the idea that *A. muciniphila* growth and metabolic activity may contribute substantially to the clinical response observed in WBF-011 recipients. Future intervention studies integrating longitudinal metagenomic and host-response data could help establish more direct mechanistic links between *A. muciniphila* growth, microbial metabolism, and host outcomes.

Building on the strong agreement observed between predicted growth and empirically-estimated engraftment in two probiotic intervention studies, we applied MCMM simulations to a larger cohort of generally healthy participants from Arivale ($N = 1,786$) who were coached to improve their diet and lifestyle. Using MCMMs built from this data set, we simulated a switch from a low- to a high-fiber diet, and compared how changes in predicted butyrate (Δbutyrate) were associated with longitudinal changes in cardiometabolic markers (LP-IR, insulin, HOMA-IR) and immune activation markers (white blood cell, neutrophil, lymphocyte counts). Arivale participants were generally guided toward a higher-fiber, butyrogenic diet, and these findings highlight the potential of MCMM-based approaches to predict personalized microbiome-mediated responses to dietary interventions.

Microbial production of butyrate and propionate increased in response to prebiotic supplementation and a shift to a standard high-fiber diet. Interestingly, addition of the WBF-011 probiotic cocktail generally led to decreases in butyrate and propionate production, which was partially explained by concomitant increases in lactate and acetate. Although the cocktail is designed to be butyrogenic, the limited growth observed for butyrate producers *A. hallii*, *C. beijerinckii,* and *C. butyricum* is consistent with this result. Individual-specific modeling, however, allows identification of the most effective treatment for each sample, in terms of SCFA production. Responses varied across individuals, with not all subjects benefiting equally from the same prebiotic fibers, and some individuals showed higher butyrate production in the presence of WBF-011, despite the average negative trend across individuals. This highlights the potential of MCMMs as a tool for personalizing prebiotic and probiotic combinations.

MCMMs have several inherent limitations. Chief among these is that MCMMs rely on simplifying steady-state assumptions that restrict their ability to fully capture the temporal dynamics of gut microbial communities. Moreover, the MCMM framework depends on the availability of accurate GEMs for all of the taxa in a sample, and missing annotations or reactions in existing GEMs can lead to inaccurate predictions. Existing GEM databases do not include all extant strains for a given gut bacterial species. The consequence of this is that MCMM-based analyses constrained by real-world gut microbiome samples are often limited to species-level resolution, using models that summarize the metabolic capacity of all available strains within a given species. Our study relied on species-level GEMs from the AGORA2 database, due to their relatively higher levels of curation and quality, compared to *de novo* constructed models [28,29]. Future work should focus on accounting for strain-level differences in metabolic capacity. Another limitation was the lack of participant dietary intake data. Better constraints on personalized dietary intake could further improve the predictive accuracy that we observed in the current analysis, which assumed that everyone was eating a standard European diet. Finally, the current methodology for constructing and solving MCMMs does not account for spatial structure or host physiology within the gut. For instance,

*A. muciniphila* typically grows within the mucus layer of the colonic epithelium, in a different microenvironment than many other taxa in the microbiota. For this reason, it is likely to be more exposed to host-derived compounds. Future work should aim to integrate MCMMs with metabolic models of the colonic epithelium [30], which may help to further improve predictions.

Taken together, these findings demonstrate the utility of MCMMs as a predictive framework for assessing prebiotic, probiotic, and dietary interventions at the individual and population levels. By integrating metabolic modeling with experimental validation, we highlight key ecological and metabolic determinants of probiotic engraftment success and SCFA production, providing a pathway for more targeted microbiome-based therapies. However, these approaches are still evolving and require further validation in prospective human trials. Realizing the full potential of these models will require continued refinement, including improved strain-level resolution, personalized dietary intake constraints, and enhanced host–microbiome interaction modeling. Future research should aim to bridge these gaps by incorporating multiomics approaches, longitudinal sampling, and personalized dietary profiling. Ultimately, leveraging MCMMs in a clinical setting could enable precision microbiome therapeutics, optimizing probiotic, prebiotic, and dietary intake to more effectively treat a wide range of acute and chronic diseases.

## Methods

### Data collection

Data from Validation Study A were obtained from Perraudeau and colleagues [24]. The study cohort consisted of 76 participants previously diagnosed with type 2 diabetes. Participants were randomized into two treatment groups ($N=27$ and $N=23$) and a placebo group ($N=26$). 6, 2, and 10 participants were lost to follow-up in each study arm, respectively. Those in the experimental groups received one of two probiotic formulations, WBF-010 or WBF-011. WBF-010 contained strains of *Bifidobacterium longum subsp. infantis*, *Clostridium butyricum*, and *Clostridium beijerinckii*. WBF-011 included the same strains as WBF-010, with the addition of *Akkermansia muciniphila* and *Anaerobutyricum hallii*. Both probiotic formulations were supplemented with a small amount of inulin (0.3 g), a prebiotic derived from chicory root. Participants in all three study arms followed the intervention for 12 weeks, consuming three capsules twice daily. The primary metabolic endpoint assessed in the original trial was the change in the area under the glucose curve (AUC) during a standard three-hour meal-tolerance test conducted at baseline and after 12 weeks. The ΔglucoseAUC values were calculated by subtracting baseline AUC from the 12-week AUC from the same individual. Analysis revealed that WBF-011, but not WBF-010, led to a significant shift in ΔglucoseAUC compared to the placebo. All data were shared directly by the authors of the original study.

Data from Validation Study B were obtained from Dsouza and colleagues [12]. This cohort consisted of 32 individuals treated with an 8-strain probiotic cocktail called VE303, containing strains of *Enterocloster bolteae*, *Anaerotruncus colihominis*, *Sellimonas intestinalis*, *Clostridium symbiosum*, *Blautia sp001304935*, *Dorea longicatena*, *Clostridium innocuum*, and *Flavonifractor plautii*. Twenty participants who were treated with antibiotics prior to probiotic administration and had follow-up data at 12 weeks were used to test the modeling approach. Of the strains included in the cocktail, 6 out of 8 (excluding *Clostridium innocuum* and *Blautia sp001304935*) were represented in the AGORA2 database [25] and could be modeled using MCMMs. Shotgun metagenomes from this study were collected from the NCBI Sequence Read Archive under the accession number PRJNA755324.

Additional data used in the analysis were collected from Arivale, (Seattle, WA). Arivale closed its operations in 2019. For analyzing the shift in butyrate production between a standard European and a high-fiber diet, V4 16S rRNA amplicon data were used for samples that had paired baseline fecal and blood sampling, as well as a secondary blood sampling between 4 and 8 months later ($N=1,786$). Participants in this analysis were adults in the United States (1,079 female, 707 male, sex assigned at birth), with an average age of $48.5 \pm 0.3$ years. 16S data for the Arivale cohort can be found on the NCBI Sequence Read Archive under accession numbers PRJNA826530 and PRJNA826648. For exploratory analysis

testing combinatoric prebiotic, probiotic, and dietary interventions, a subset of these participants for whom shotgun metagenomic sequencing was available was used ($N=156$). De-identified metagenomic data were collected and used to construct MCMMs as described below. Participants included in this sub-cohort analysis were adults in the United States (92 females and 64 males, sex assigned at birth), with an average age of $47.4 \pm 1.0$ years. Arivale metagenomes were deposited at the NCBI Sequence Read Archive under accession number PRJNA1262070. Further details on the Arivale cohort are described in Zubair and colleagues [31].

## qPCR data processing

qPCR data from Validation Study A were used as ground truth for validating model-based predictions. These data were provided by the authors of the original study. Custom primer pairs were designed for each of the five probiotic strains in the WBF-011 treatment [24]. Cycle threshold (Ct) values were obtained for all five species at baseline (Week 0), during treatment (Week 4), and at the end of treatment (Week 12). Samples failing quality control were excluded from analysis. Despite using strain-specific primer pairs, all five probiotic strains were detected in at least one sample at Week 0, suggesting cross-reactivity with endogenous strains.

Because qPCR standard and melt curves from the original experiments were not retrievable, the resulting data were semiquantitative and were used to assess within-species trends, rather than estimate absolute abundances. −Ct was used as a proxy for relative species abundance. For each species, an engraftment threshold was defined as the mean −Ct at Week 0, representing the baseline amplification level. Week 12 values were binarized relative to this threshold to classify each sample as either exhibiting growth or non-growth.

## Shotgun metagenome data processing

For Validation Study B, metagenomic data from Week 0 and Week 12 were used as ground truth to validate model predictions. Metagenomic data from Week 0 were used for both Validation Study A and Validation Study B to construct MCMMs. FASTQ files were collected from the authors for Validation Study A and from the NCBI Sequence Read Archive for Study B. Raw data were adapter-trimmed and quality-filtered using fastp [32]. Quality-filtered reads were processed using Kraken2 (v1.01) for taxonomic classification, and Bracken was used to refine taxonomic abundance estimates [33,34]. The Kraken2 default database (based on Refseq release 94) was used. Species with fewer than 10 assigned reads were omitted. The analysis pipeline is available at https://github.com/Gibbons-Lab/pipelines/tree/master/metagenomics. The same metagenomic processing pipeline was run on the shotgun metagenomic data for Arivale participants.

For Validation Study B, species relative abundances from Week 12 were used as a benchmark for comparison with model predictions. An increase in relative abundance of 50% was used for binarization, subject to an additional relative abundance threshold filter of 0.005 at Week 12. Using this method, each sample/species pair was classified as showing growth or non-growth.

## 16S rRNA amplicon data processing

16S rRNA amplicon sequencing data were processed as described by Diener and colleagues [35]). Briefly, reads were processed using DADA2 to infer amplicon sequence variants, learn run-specific error profiles, merge counts, and remove chimeras [36]. Taxonomy was assigned using a naive Bayes classifier and the Silva database (v128) [37]. Approximately 90% of reads were classified to genus level and 32% to species level. Low-abundance taxa present in fewer than 50% of samples or with an average of less than 10 reads per sample were excluded.

## Model construction

Read counts were normalized to relative abundance before their incorporation into downstream modeling workflows. MCMMs were constrained using 16S amplicon or metagenomic relative abundance profiles for all baseline samples

using MICOM (v0.37.0), a framework for simulating microbial community-scale metabolism [18]. Taxonomic profiles from baseline samples were mapped to GEMs from the AGORA database (v2.01) [25], which were used to construct community-scale metabolic models reflecting the microbial composition of each individual microbiome. A lower relative abundance threshold of 0.001 was applied during model construction to exclude taxa representing less than 0.1% of the community composition.

For samples in Validation Studies A and B, probiotic supplementation was simulated *in silico* by modifying the baseline taxonomic composition. For Validation Study A, the relative abundances of endogenous taxa were proportionally reduced to 80% of their original values, while five probiotic species from WBF-011 were introduced at a total relative abundance of 20% (4% each). For Validation Study B, the relative abundances of endogenous taxa were proportionally reduced to 20% of their original values to reflect vancomycin pre-treatment, while 6 probiotic species from VE303 cocktail were introduced at a total relative abundance of 80% (13.33% each). These models were constructed at the species level.

For the dietary intervention modeling, MCMMs were constructed from 16S rRNA amplicon sequencing data available for 1,786 samples in the Arivale cohort with paired stool and blood samples at baseline and a blood sample from a 4- to 8-month follow-up sampling. Due to the taxonomic resolution of 16S rRNA amplicon sequencing data, these models were constructed at the genus level.

For the additional analysis of prebiotic, probiotic, and dietary combinations in the Arivale cohort, MCMMs were constructed at the species-level using metagenomic sequencing data available for 156 samples. Probiotic administration was simulated by scaling relative abundances of endogenous taxa to 80% of their original values, and introducing 20% relative abundance of the probiotic cocktail from Study A (4% each). These models were constructed at the species level.

## Model growth

Community growth simulations were performed after selecting MICOM's cooperative tradeoff parameter using the micom.workflows.grow() workflow, which balances individual taxon growth with overall community-wide metabolic efficiency. A tradeoff parameter of 0.99 was applied because it was the closest value to 1.0 that also allowed for >80% of taxa to grow in Validation Study A. This parameter value was then fixed and applied to all other cohorts in this study to avoid overfitting. For Validation Studies A and B, models were supplied with a simulated growth medium reflective of an average European diet collected from the Virtual Metabolic Human diet database (https://www.vmh.life/#nutrition). Metabolites in the medium that can be taken up by the host, as defined by uptake reactions in the Recon3D model, were reduced to 20% of their original flux to account for absorption in the small intestine [38]. Additionally, host-derived compounds such as mucins and bile acids that were missing from the medium were added. The medium was further supplemented with a minimal set of metabolites necessary to support the growth of all taxa in the model database, using MICOM's complete_db_medium() function. In the absence of detailed dietary recall data, this can be used to approximate the average dietary consumption of individuals in North America, where the cohorts included in this study were located. For Validation Study A, 0.3 g or 30 g of inulin was included as an additional substrate in the medium to account for the prebiotic component of the synbiotic intervention. Predicted growth rates were collected from the output of the micom.workflows.grow() workflow, and predicted metabolic production rates were calculated using the micom.measures.production_rates() function in MICOM.

For analysis of the Arivale cohort, simulations were conducted using a standard European diet as well as a high-fiber diet rich in resistant starch. This media was also collected from the Virtual Metabolic Human database, and augmented in the same fashion as the standard European medium. The same cooperative tradeoff parameter as in Validation Studies A and B was used. Metabolic production from the model output was calculated using the micom.measures.production_rates() function in MICOM. Δbutyrate was calculated as predicted microbial butyrate production on a standard European diet, subtracted from predicted butyrate production on a high-fiber diet. Changes in clinical chemistry values (Δchemistries) were calculated as the value at enrollment subtracted from the value at follow-up (4–8 months post enrollment).

These values were used as the independent and dependent variables for regression analysis, respectively (see Statistical analysis below).

Inulin, pectin, resistant starch, resistant maltodextrin, cellulose (hemp seed), or arabinoxylan (psyllium husk) were added to the standard European diet to simulate the addition of various prebiotics. All prebiotic additions were added in a manner that balanced the overall carbon content to enable comparison between interventions, equivalent to a prebiotic supplementation of approximately 30 g per day. Predicted growth rates for the five probiotic species from Validation Study A, MICOM model outputs, and predicted metabolic production rates were calculated using the micom.measures.production_rates() function. To facilitate direct comparison with engraftment scores in the validation studies, predicted growth rates were binarized using a threshold of $10^{-2}$ $h^{-1}$ (corresponding to a minimal doubling time of 69.3 h). Growth rates below this value were classified as non-engraftment (0), while those above were classified as successful engraftment (1). Sensitivity of the results of Validation Studies A and B to this growth rate threshold was tested (S1 and S2 Figs).

## Statistical analysis

All statistical analysis conducted in this study was done using Scipy (v1.13.1) [39], scikit-learn (v1.6.1) [40], or statsmodels (v0.14.4) [41].

To assess significant increases in abundance of probiotic species from qPCR data, paired *t*-tests were used to compare Week 0 and Week 12 −Ct values. Statistical analyses were performed in Python using the scipy.stats.ttest_rel() function. In instances where only a single −Ct value was available from the Week 0 time point, a one-sample *t* test was used to compare that sample to the 12-week time point using the scipy.stats.ttest_1samp() function.

To assess the concordance between binarized MICOM-predicted engraftment and qPCR observations, Cohen's Kappa statistic was computed using sklearn.metrics.cohen_kappa_score(). This measure quantifies the level of agreement beyond what would be expected by chance, providing a stringent estimate of the predictive accuracy of MCMMs for probiotic engraftment outcomes. Furthermore, Fisher's exact test was used to evaluate whether the observed agreement between MICOM predictions and qPCR results in the confusion matrix was statistically significant, using the scipy.stats.fisher_exact() function.

Significance differences in predicted butyrate and propionate between treatment groups in both the probiotic trial and the Arivale cohort (Figs 3A, 3B, and 5A–5D) were assessed using a non-parametric Mann-Whitney U test, using the scipy.stats.mannwhitneyu() function. Significance differences between predicted butyrate and propionate between optimal and standard of care interventions in the Arivale cohort (Fig 5F and 5G) were also determined using non-parametric Mann–Whitney *U* tests.

Significant differences in Δglucose AUC and *A. muciniphila* growth rates between treatment groups were determined using non-parametric Mann–Whitney *U* test, using the scipy.stats.mannwhitneyu() function. The association between *A. muciniphila* growth rate and change in glucose AUC was assessed using ordinary least squares linear regression, using the statsmodel.api.ols() function. In the absence of available metadata, no covariates were used.

Regression analysis associating changes in butyrate production between a standard European and high-fiber diet to changes in a set of 127 blood-based clinical chemistries (Fig 4) was conducted using statsmodels, with age, sex and the baseline values of clinical chemistries as covariates, using the statsmodel.api.ols() function. Correction for multiple comparisons was performed using the Benjamini–Hochberg method[42], using the statsmodels.stats.multitest.fdrcorrection() function.

## Ethics statement

Data from Validation Study A were obtained directly from the original trial investigators under a confidentiality agreement. The trial protocol was approved by the Central Institutional Review Board (Allendale, Old Lyme, Connecticut, USA),

and all participants provided written informed consent prior to enrollment. The study is registered at ClinicalTrials.gov (NCT03893422).

The human participant data for Validation Study B are publicly available and were collected from the NCBI SRA under accession number PRJNA755324. The study was approved by the IntegReview Institutional Review Board (Austin, Texas, USA) and monitored by the U.S. Food and Drug Administration and a safety review committee. Written informed consent was obtained from all participants prior to enrollment. The study is registered at ClinicalTrials.gov under trial identifier NCT04236778.

All procedures for the Arivale cohort study were reviewed and approved by the Western Institutional Review Board (WIRB) (IRB study numbers 20170658 at the Institute for Systems Biology and 1178906 at Arivale). The research was conducted using fully deidentified and aggregated human participant data from individuals who had signed a research authorization permitting the use of their anonymized data for research purposes. In accordance with current U.S. regulations governing the use of deidentified data, informed consent was not required for this analysis.

## Supporting information

**S1 Fig. Sensitivity of Validation Study A to growth rate threshold.** Agreement fraction (red line) and Cohen's $\kappa$ (blue line) are shown as a function of the growth rate threshold, around which model predictions are binarized for growth or non-growth. A black vertical line is shown at 0.01, the value used in this analysis, equivalent to doubling time of ~70 hours. Underlying available in S2 Data.
(TIFF)

**S2 Fig. Sensitivity of Validation Study B to growth rate threshold.** Agreement fraction (red line) and Cohen's $\kappa$ (blue line) are shown as a function of the growth rate threshold, around which model predictions are binarized for growth or non-growth. A black vertical line is shown at 0.01, the value used in this analysis, equivalent to doubling time of ~70 hours. Underlying available in S2 Data.
(TIFF)

**S3 Fig. Scaling metabolic predictions by community growth rate.** To ensure that observed shifts in metabolite production were not driven by overall changes in biomass, production rates were normalized by predicted community growth rates, which reflect the total biomass produced by the community. Scaled values for SCFA and lactate production were largely consistent with the unscaled predictions across prebiotic and probiotic conditions. Significant differences between ±probiotic treatments within each prebiotic or diet intervention were assessed using a Mann–Whitney $U$ test ($p < 0.05$; $p < 0.01$; $p < 0.001$). Underlying available in S5 Data.
(TIFF)

**S4 Fig. Individual-specific optimal prebiotic, probiotic, and synbiotic treatments for maximizing butyrate production.** MCMM-predicted butyrate production varied substantially across individuals ($N = 156$) in response to different prebiotic and probiotic combinations. The combination of psyllium husk with no probiotic cocktail produced the highest butyrate flux in the greatest number of individuals. However, every prebiotic/probiotic combination was optimal for at least one individual, with the exceptions of: (1) hemp seed, and (2) the no-prebiotic condition. Heatmap colors indicate predicted butyrate flux (mmol/gDW/h), and black boxes denote the most effective treatment for each sample. Underlying available in S5 Data.
(TIFF)

**S5 Fig. Individual-specific optimal prebiotic, probiotic, and synbiotic treatments for maximizing propionate production.** MCMM-predicted propionate production also varied substantially across individuals ($N = 156$) in response

to different prebiotic and probiotic combinations, though slightly less than for butyrate. The combination of psyllium husk without a probiotic cocktail produced the highest propionate flux in the greatest number of individuals. However, every other prebiotic/probiotic combination was optimal for at least one individual, with the exceptions of: (1) hemp seed, and (2) the no-prebiotic condition. Heatmap colors indicate predicted propionate flux (mmol/gDW/h), and black boxes denote the most effective treatment for each sample. Underlying available in S5 Data.
(TIFF)

**S1 Data. qPCR and metagenomic detection data for Validation Studies A and B, corresponding to Fig 1.**
(XLSX)

**S2 Data. MCMM predictions, ground truth values, and agreement metrics for Validation Studies A and B, corresponding to Fig 2.**
(XLSX)

**S3 Data. SCFA production, glucose AUC, and *Akkermansia muciniphila* growth for Validation Study A, corresponding to Fig 3.**
(XLSX)

**S4 Data. SCFA production, clinical laboratory measurements, and regression results for the Arivale cohort, corresponding to Fig 4.**
(XLSX)

**S5 Data. SCFA production, engraftment results, and optimal treatment predictions for the Arivale cohort, corresponding to Fig 5.**
(XLSX)

## Acknowledgments

Thanks to Christian Diener and members of the Gibbons Lab for useful discussions.

## Author contributions

**Conceptualization:** Nick Quinn-Bohmann, Alex V. Carr, Sean M. Gibbons.

**Data curation:** Nick Quinn-Bohmann.

**Formal analysis:** Nick Quinn-Bohmann, Alex V. Carr.

**Funding acquisition:** Sean M. Gibbons.

**Investigation:** Nick Quinn-Bohmann, Sean M. Gibbons.

**Methodology:** Nick Quinn-Bohmann, Alex V. Carr.

**Project administration:** Sean M. Gibbons.

**Supervision:** Sean M. Gibbons.

**Validation:** Nick Quinn-Bohmann, Alex V. Carr.

**Visualization:** Nick Quinn-Bohmann.

**Writing – original draft:** Nick Quinn-Bohmann.

**Writing – review & editing:** Nick Quinn-Bohmann, Alex V. Carr, Sean M. Gibbons.

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
