## [Editor Report · Decision Letter 0]

30 May 2025

Dear Dr Gibbons,

Thank you for submitting your manuscript entitled "Metabolic modeling reveals determinants of synbiotic efficacy in a human intervention trial" for consideration as a Research Article by PLOS Biology.

Your manuscript has now been evaluated by the PLOS Biology editorial staff and I am writing to let you know that we would like to send your submission out for external peer review.

Once your full submission is complete, your paper will undergo a series of checks in preparation for peer review. After your manuscript has passed the checks it will be sent out for review. To provide the metadata for your submission, please Login to Editorial Manager (https://www.editorialmanager.com/pbiology) within two working days, i.e. by Jun 10 2025 11:59PM.

Kind regards,

Luke

Lucas Smith, Ph.D.

Senior Editor

PLOS Biology

lsmith@plos.org

---

## [Decision Letter · Decision Letter 1]

31 Jul 2025

Dear Sean,

Thank you for your patience while your manuscript "Metabolic modeling reveals determinants of synbiotic efficacy in a human intervention trial" was peer-reviewed at PLOS Biology and I apologize again for the protracted review process. Your study has now been evaluated by the PLOS Biology editors, an Academic Editor with relevant expertise, and by two independent reviewers. As a note, we actually had a third reviewer signed on to help assess the study, but it appears that they have dropped off. Fortunately, we think the two reviewers who have submitted their comments cover the relevant expertise, and after discussing their feedback with the Academic Editor, we have decided to move forward based on the feedback we have on hand. If the last reviewer ends up sending in their comments in the next week or so, we will forward you their feedback as well.

In light of the reviews, which you will find at the end of this email, we would like to invite you to revise the work to thoroughly address the reviewers' reports.

As you will see below, the reviewers are fairly positive about the study, and its potential interest to the field, however each reviewer has raised suggestions to strengthen the study further. We think that the reviewer comments should be thoroughly addressed before we can consider your study for publication. While many of the reviewer comments are addressable with textual changes, we think it will be important for you to develop the study further by providing the additional validation studies and analyses requested by the reviewers.

Given the extent of revision needed, we cannot make a decision about publication until we have seen the revised manuscript and your response to the reviewers' comments. Your revised manuscript is likely to be sent for further evaluation by all or a subset of the reviewers.

**IMPORTANT - SUBMITTING YOUR REVISION**

*Re-submission Checklist*

*Published Peer Review*

*PLOS Data Policy*

*Blot and Gel Data Policy*

Sincerely,

Luke

Lucas Smith, Ph.D.

Senior Editor

PLOS Biology

lsmith@plos.org

REVIEWS:

Reviewer's Responses to Questions

Reviewer #1: This study explores how microbial community-scale metabolic models (MCMMs) can predict the success of synbiotic interventions (combinations of probiotics and prebiotics) in human subjects. The authors used data from a placebo-controlled trial where participants received a synbiotic cocktail (WBF-011) containing five probiotic strains and inulin, a prebiotic fiber. The authors highlight the potential of MCMMs as tools for precision microbiome therapeutics, capable of customizing interventions based on an individual's microbiome composition and dietary background. This manuscript presents an innovative and timely contribution to microbiome research and precision health. With more rigorous discussion of modeling limitations, clarification of the clinical relevance of findings, and restrained interpretation of personalization claims, this work will be a valuable addition to the literature.

Comments:

While the authors report >85% accuracy for predicting probiotic engraftment using MCMMs, the extent of external validation is limited. Given the complexity and heterogeneity of gut microbiota and host factors, it is critical to test the model's robustness across independent cohorts or datasets. The use of a secondary observational cohort is a good start but lacks the depth of validation necessary for broader translational claims. Please include additional validation analyses, or clearly state the exploratory nature of the personalization findings and their limitations.

MCMMs rely on several simplifying assumptions (e.g., steady-state flux balance, known pathways, representative genomes) that limit their ability to capture host-specific microbiome dynamics. The discussion section does not adequately address these model constraints. Include a more nuanced discussion of model limitations and assumptions, and their potential impact on prediction accuracy and biological interpretation.

The role of diet in modulating SCFA production and probiotic colonization is critical, yet the manuscript provides little information on participants' dietary intake or control during the intervention. Please clarify whether dietary data were collected and how variation in intake was accounted for in the models and analysis.

The use of MCMMs to propose personalized synbiotic formulations is promising but currently remains theoretical. There is a risk of overstating real-world readiness. We should acknowledge this as a proof-of-concept and clearly delineate what further steps would be required to operationalize personalization clinically.

Please pay attention to spelling and grammar issues. For example, in line 75, "date" should be spelled "data".

The authors use qPCR but the results are not quantitative outside of using the inverse of the cycle threshold. While this is used as a proxy, the value of this approach is questionable. As the data is presented, the authors are using PCR which is semiquantitative.

When performing qPCR, a standard curve should be run so the results will provide a quantitative amount. Furthermore, when using something like SYBR green, the melt curve of samples should match the standard to give confidence in target detection. This would provide additional data as it relates to the comment in lines 101-103. While the authors provide a threshold for some low abundance bacteria, I don't think there is any information on how this was generated or validation data provided.

What is the difference between detection threshold and engraftment threshold? (lines 96-7 vs 112)

You provide predicted values for butyrate and propionate production from models for groups 1-4, but not group 5 (where only statistics are provided) - please provide the data.

Furthermore, detection of the species does not equate to engraftment. Detection over time does suggest that there is colonization.

The github repository for codes used in this work seems to be inaccessible.

------------------

Reviewer #2, Christoph Kaleta (note, reviewer 2 has signed this review): The study by Quinn-Bohnmann & Gibbons shows how to use metabolic modeling to predict engraftment efficiency of probiotics by the supplementation of prebiotics. They are able to predict 85% of engraftments outcomes in a synbiotic study of diabetes patients - an impressive accuracy, given the limited amount of information integrated in the modeling approach. Further, they use the metabolic modeling framework to predict SCFA production in the microbial communities and how this correlates with the systemic inflammation marker CRP in the cohort. Finally, they explore engraftment predictions in another larger cohort using different prebiotic treatments and explore how an personalized synbiotic treatment could look like, if the goal is to increase SCFA production.

I think this is an elegant study using metabolic modeling to predict synbiotic efficiency in a simple cohort design. It shows the power of metabolic modeling in clinical contexts. However, I feel that the second exploratory part with the second cohort is a little detached from the initial simulations and, though interesting as a case study, with little experimental support and could be shortened and better discussed. Other than that, I found some unclear points in methodology and have made some comments to improve the extent of data analysis, to fully understand the results.

Major points:

Fig1: Sample size in the plot looks odd, especially at the first time point where often only one or no sample was available. Why is that, can you explain? How were statistics derived for these cases?

Fig4: Why was the same analysis not applied to AUC values of glucose levels, the other endpoint measurements in the study of Perraudeau et al. (especially, after discussing this relationship in l. 261ff)? Were there other measures which have been tested in Perreadeau et al. which might be of interest in this study?

Fig5 and Fig6: The results presented are interesting, however I wonder, whether general trends are explained by two factors: first the amount of additional carbon atoms provided to the different communities, secondly the overall growth rates of the communities. For the first factor, it looks to me that independent of what is added to the community, it will increase the production of SCFA. That overestimates the specific effect of SCFA production after addition of the compounds and that the main effect is simply the addition of carbon sources. Hence, I would like to see the effect of the specific metabolite addition compared to glucose equivalents - as this would attribute the SCFA production to the specific metabolite and not to increased carbon sources (In this regard, I think hemp seeds are not metabolized at all by the community, comparing it to no prebiotics - could you check, please?). For the second factor, I would argue that it is interesting to see how much more SCFA are produced per unit biomass produced - as one could argue that this ratio is more relevant than overall production. Generally, increased biomass production will increase fluxes of all other by-

products. However, it is not clear what effectively is more relevant in physiological conditions, yet I think the analysis is easily done and gives some more insights in the mechanisms at work.

Minor

l86-89: I would suggest to shorten to "Microbial abundances for focal species were determined using qPCR (see Method)" to increase clarity of the text.

l106-108: "... metagenomic sequencing data from participants in the WBF-011 treatment arm to assess whether MCMMs could predict the engraftment patterns... " this sentence implies that new metabolic models were reconstructed using the metagenomic data. However, from the methods it reads, that the data was "only" matched to AGORA(2) models. Could you please clarify, what has been done exactly and adjust wording please?

l110-122/Fig2: Could you provide a sensitivity analysis of the results for the given threshold of 1e-3 for growth in the simulation, please? This would give insights on how important it is to choose the correct threshold and how prone the analysis is to overfitting.

Fig3: I do not quite understand why there is a distinction between the placebo and treatment group as the analysis uses metabolic modeling to explore the effects of pro-, pre- and synbiotics. I would recommend either to remove the placebo group or combine both groups. However, this is more a narrative flaw than a technical or scientific one.

FIg5A: I would suggest it would be more informative to provide a bar chart of percentage of samples engrafted per condition and bacterium, instead of the "heatmap".

l313-321: you discuss the shortcoming of missing strain level resolution due to lack of resolution in the databases and advice to resolve these in the future. I think it is advisable to mention here metabolic modeling reconstruction frameworks, which could be used to derive strain level metabolic models, like pathway tools, CarveMe or gapseq.

l394: "A tradeoff parameter of 0.99 was applied..." this is a very high tradeoff parameter, when compared to results of the original paper introducing cooperative tradeoff metabolic modeling. Could you please explain, why this high trade-off parameter was used, instead of something recommended like 0.5-0.7?

---

## [Editor Report · Decision Letter 2]

17 Dec 2025

Dear Sean,

Thank you for your patience while we considered your revised manuscript "Metabolic modeling reveals determinants of prebiotic and probiotic treatment efficacy across two human intervention trials" for publication as a Research Article at PLOS Biology. This revised version of your manuscript has been evaluated by the PLOS Biology editors and the Academic Editor who is fully satisfied by the revision and thinks you have done a good job addressing the previous reviewer comments.

Based on our Academic Editor's assessment of your revision, we are likely to accept this manuscript for publication. However, before we can do so, we need you to address a number of data and other policy-related requests in a last revision. These requests are detailed below.

**IMPORTANT - Please address the following editorial requests:

1) ABSTRACT: Please note that per journal policy, the model system/species studied should be clearly stated in the abstract of your manuscript. Please update the abstract to explicitly indicate that your study examined data from human cohorts.

2) FINANCIAL DISCLOSURES STATEMENT: Please update your financial disclosures statement, in our editorial manager system, to indicate whether the sponsors or funders played any role in the study design, data collection and analysis, decision to publish, or preparation of the manuscript. (I see you did this in the competing interests section, but we need it in the financial disclosures statement as well).

3) ETHICS STATEMENT: I see that you have indicated that your study does not require an ethics statement because you re-analyze data from other studies. For studies where the anonymized data is publicly available, I think you may be correct, but we ask that you add details to your methods section, documenting the original study's ethics status (reporting whether those studies were conducted with informed written consent, under the principals of the declaration of Helsinki, etc). For the analyses of data that is not publicly available, we think your study likely does require approval, and ask that you provide an updated ethics statement that includes the following details:

-- Please include the full name of the IACUC/ethics committee that reviewed and approved the animal care and use protocol/permit/project license. Please also include an approval number.

-- Please include the specific national or international regulations/guidelines to which your animal care and use protocol adhered. Please note that institutional or accreditation organization guidelines (such as AAALAC) do not meet this requirement.

-- Please include information about the form of consent (written/oral) given for research involving human participants. All research involving human participants must have been approved by the authors' Institutional Review Board (IRB) or an equivalent committee, and must have been conducted according to the principles expressed in the Declaration of Helsinki.

If your study is considered exempt from requiring ethical approval, please indicate which ethical body deemed it exempt and add a sentence to the methods section explaining why ethical approval is not needed in this case.

4) DATA: We understand that some of the data examined in your study is not publicly available, but rather was provided by the authors of Perraudeau et al. We ask that you please:

a. In the editorial manager system, update your data availability statement to provide a more complete description of the data set and the third-party source. Please be sure to provide all necessary contact information that others would need to apply to gain access to the data. Ideally, this would be a data access committee, ethics committee, or other institutional body, as this is more robust than directing researchers to a single author.

b. Please provide as much underlying data as you are legally able to distribute. We understand that you may not be allowed to share the raw data from Perraudeau et al. If possible, however, we would ask that you share the processed data underlying your figures.

For example, if possible, we ask that you provide the individual quantitative observations that underlie the data summarized in the figures and results of your paper, as a supplementary files (e.g., excel). Please ensure that all data files are uploaded as 'Supporting Information' and are invariably referred to (in the manuscript, figure legends, and the Description field when uploading your files) using the following format verbatim: S1 Data, S2 Data, etc. Multiple panels of a single or even several figures can be included as multiple sheets in one excel file that is saved using exactly the following convention: S1_Data.xlsx (using an underscore).

5) CODE: Thank you for providing a github link to access the code used in your study. Please note that we cannot accept sole deposition of code in GitHub, as this could be changed after publication. We therefore ask that you also archive this version of your publicly available GitHub code to Zenodo. Once you do this, it will generate a DOI number, which you will need to provide in the Data Accessibility Statement (you are welcome to also provide the GitHub access information). See the process for doing this here: https://docs.github.com/en/repositories/archiving-a-github-repository/referencing-and-citing-content

We expect to receive your revised manuscript within three weeks.

*Published Peer Review History*

*Press*

Sincerely,

Luke

Lucas Smith, Ph.D.

Senior Editor

lsmith@plos.org

PLOS Biology

---

## [Editor Report · Decision Letter 3]

21 Jan 2026

Dear Sean,

Thank you for the submission of your revised Research Article "Metabolic modeling reveals determinants of prebiotic and probiotic treatment efficacy across multiple human intervention trials" for publication in PLOS Biology, and thank you for addressing our editorial requests in this revision. On behalf of my colleagues and the Academic Editor, Ran Blekhman, I am pleased to say that we can in principle accept your manuscript for publication, provided you address any remaining formatting and reporting issues. These will be detailed in an email you should receive within 2-3 business days from our colleagues in the journal operations team; no action is required from you until then. Please note that we will not be able to formally accept your manuscript and schedule it for publication until you have completed any requested changes.

PRESS

We frequently collaborate with press offices. If your institution or institutions have a press office, please notify them about your upcoming paper at this point, to enable them to help maximize its impact. If the press office is planning to promote your findings, we would be grateful if they could coordinate with biologypress@plos.org. If you have previously opted in to the early version process, we ask that you notify us immediately of any press plans so that we may opt out on your behalf.

Sincerely,

Lucas Smith, Ph.D.

Senior Editor

PLOS Biology

lsmith@plos.org